# High Prevalence of Astigmatism in Children after School Suspension during the COVID-19 Pandemic Is Associated with Axial Elongation

**DOI:** 10.3390/children9060919

**Published:** 2022-06-19

**Authors:** Suei-Cheng Wong, Chea-Su Kee, Tsz-Wing Leung

**Affiliations:** 1Centre for Eye and Vision Research (CEVR), 17W Hong Kong Science Park, Hong Kong; sc.wong@cevr.hk (S.-C.W.); c.kee@polyu.edu.hk (C.-S.K.); 2School of Optometry, The Hong Kong Polytechnic University, Hung Hom, Kowloon, Hong Kong; 3Research Centre for SHARP Vision (RCSV), The Hong Kong Polytechnic University, Kowloon, Hong Kong

**Keywords:** astigmatism, children, refractive error, COVID-19

## Abstract

During the COVID-19 pandemic, the Hong Kong Government enforced a “school from home” policy between February and September 2020. This cross-sectional epidemiological study was conducted to investigate the prevalence of astigmatism and visual habits after the home confinement period. Vision screenings were conducted at three local government-funded primary schools in Hong Kong from October 2020 to December 2020. A total of 418 ethnically Chinese primary school children completed the eye examination and returned questionnaires concerning demographic information and visual habits. It was found that 46.5% (95% CI, 41.7–61.4%) of the children aged 8 to 11 years had astigmatism ≥ 0.75 D, which was predominately With-The-Rule astigmatism. The prevalence of astigmatism reported in these children is generally higher than that of studies conducted before COVID. Compared to their non-astigmatic peers, astigmatic children had a longer axial length (*p* < 0.001) and engaged in fewer outdoor activities (*p* = 0.04). Multiple linear regression analyses also revealed significant relationships between axial length and both cylindrical error and *J*0 astigmatism. Due to the high astigmatism prevalence, there is a pressing need for further studies on the long-term impact of the pandemic on children’s vision.

## 1. Introduction

Refractive (or manifest) astigmatism is a common refractive state that affects 14.9% (≥0.75 DC) of children worldwide [1]. Unlike myopia (short-sightedness) and hyperopia (long-sightedness), astigmatism imposes optical blur on the retina that cannot be alleviated by adjusting the viewing distance or ocular accommodation. The degraded retinal optical quality can only be corrected by wearing ophthalmic aids or undergoing refractive surgery. The prevalence of astigmatism varies substantially across ages [2], geographic regions [1], and ethnicities [3,4,5]. Even in the Asian Chinese population, the prevalence of astigmatism (≥0.75 DC) varies across geographic regions (i.e., countries or provinces) and places of residence (i.e., urban or rural), ranging from 9.5% to 34.98% [6,7,8,9,10,11,12]. While the etiology of astigmatism is still unclear, a recent meta-analysis of the prevalence of refractive error in Chinese children has indicated that astigmatism is more prevalent in urban than rural areas [13]. Thus, it is legitimate to hypothesize that environmental factors may promote the development of astigmatism in children.

Social isolation during COVID-19 has drastically changed people’s lifestyles. In Hong Kong, COVID was first documented in January 2020. To contain the spread of the virus, the Hong Kong Government tightened social distancing measures by closing non-essential businesses, restricting dine-in services, and prohibiting group gatherings. All face-to-face teaching in schools was suspended sporadically over the year, specifically from 29 January to 27 May, 13 July to 23 September, and 2 to 31 December. Because of the school suspensions, Hong Kong primary school children attended class in-person for less than 90 days in 2020. They were forced to stay at home resulting in changes to their visual habits, specifically more digital device usage and fewer outdoor activities [14]. Recent epidemiological studies have consistently indicated that it is likely that due to these lifestyle changes, myopia in Chinese children progressed faster during COVID, with the prevalence increasing by 7.6% to 34.3% [15,16]. However, trends in astigmatism development during the COVID pandemic have been hardly investigated.

Based on a single Hong Kong local primary school, our recent study revealed a 1.5-fold increase in the proportion of astigmatism (≥0.75 DC), from 35.4% in 2018 (before COVID) to 56.6% in 2020 (during COVID) [17]. However, whether and how visual habits contributed to the rise in the prevalence of astigmatism and whether astigmatism is associated with axial myopia remains largely unclear. The current epidemiological study extended our previous study by including a larger sample size from three local schools and examined the risk factors associated with astigmatism in Hong Kong Chinese children during the COVID-19 pandemic.

## 2. Methods

### 2.1. Study Population

All data were collected during the school reopening period between October to December 2020, which was 8 to 10 months after the first school suspension in Hong Kong to contain the COVID virus. A total of 949 schoolchildren from three government-funded local primary schools were invited to participate in this study, of whom 662 participated in the vision screening (participation rate = 69%), and 467 returned the questionnaires. These three local primary schools were located in central areas of Hong Kong (2 in Kowloon and 1 in North Hong Kong Island), where the populations are most dense. Forty-eight participants were excluded due to non-Chinese ethnicity (*n* = 2), age older than 11 years (*n* = 4), incomplete data (*n* = 10), and receiving orthokeratology treatment (*n* = 32). Before the vision screening started, informed written consent was obtained from parents after explaining the nature and purpose of the study via written notice from the school. The study followed the tenets of the Declaration of Helsinki and was approved by the human ethics committee of The Hong Kong Polytechnic University (HSEARS20190625001).

### 2.2. Vision Screening Procedure

All vision screenings were conducted at the school campus during regular school hours, from 9 am to 1 pm. Participants from all three schools underwent the same procedures of eye examination using the same set of ophthalmic instruments. Their monocular habitual (corrected or uncorrected) distance visual acuity was obtained using the Early Treatment Diabetic Retinopathy Study acuity chart (Precision Vision, La Salle, IL, USA) at 4 m. Non-cycloplegic autorefraction was measured by an open-field auto-refractor (Shin-Nippon, NVision-K 5000, Tokyo, Japan) with the fixation target, a Maltese cross, located 6 m away from the eyes. Five consecutive measurements were taken, and refractive errors generated by the instrument were analyzed. Axial length was measured by a non-contact optical biometer using partial coherent interferometry (IOL Master 500, Carl Zeiss Meditec, Jena, Germany). The averaged value of five consecutive readings with a signal-to-noise ratio >2 was used for analysis. Both an open-field auto-refractor and ocular biometer were calibrated daily before the commencement of the vision screening.

A validated, self-administered questionnaire [18] was delivered to the parents and collected by the teachers (the questionnaire was written in traditional Chinese, see the (Appendix A)for questions relevant to the current study). The questionnaire covered the children’s demographic information, family history of myopia, and visual habits (non-screen near-work, handheld digital screen work, and outdoor time) during non-school hours in the past months. Non-screen near work included all printed materials (reading, writing, and drawing), while handheld digital screen time was the sum of hours spent on tablets, computers, and smartphones. Outdoor activities included both physical and leisure activities. The average time spent on each visual task was calculated as (weekdays hours × 5 + weekend hours × 2)/7. Parental myopia was counted when at least one parent had myopia.

### 2.3. Data Analysis

Because of the high correlations of refractive errors and axial length between the right and left eyes (Pearson’s correlations, all r ≥ 0.74, *p* < 0.001), only data from the right eye were used in the analysis. Refractive errors were converted into spherical-equivalent refractive errors (SER) and *J*0 and *J*45 astigmatic components using the Fourier analysis [19].
(1)$$SER=S+\frac{C}{2}$$
(2)$$J0=\frac{-C\times cos\left(2a\right)}{2}$$
(3)$$J45=\frac{-C\times sin\left(2a\right)}{2}$$
where *S* and *C* are the spherical and cylindrical errors, respectively, and *a* represents the astigmatic axis in the negative sphero-cylindrical form. In a clinical notation, positive *J*0 indicates With-The-Rule (WTR) astigmatism, while negative *J*0 indicates Against-The-Rule (ATR) astigmatism. Positive and negative *J*45 astigmatism represents oblique astigmatism at 45° and 135°, respectively.

Astigmatism was defined as cylinder error (Cyl) ≥ 0.75 D. Other definitions (Cyl ≥ 0.50 D & ≥1.00 D) were also employed for comparison with other studies previously conducted on the Asian Chinese population. Astigmatism was further categorized into three common clinical subtypes according to the astigmatic axis—With-The-Rule (WTR) astigmatism: axis 0°–15° or 165°–180°; Against-The-Rule (ATR) astigmatism: axis 75°–105°; Oblique (OBL) astigmatism: axis 16°–74° or 106°–164°. Since non-cycloplegic refraction was performed, axial length was used when examining the relationship between myopia and astigmatism.

Data analysis was performed using SPSS Statistics (version 28.0, IBM Corp., New York, NY, USA). The significance level was set at α < 0.05. the one-way ANOVA test and independent *t*-test were used to examine the differences in ocular parameters across groups and visual habits between two groups, respectively. The chi-squared test was used to investigate the distributions of categorical variables. Multiple linear regression analyses were used to identify the risk factors associated with astigmatism, with axial length, gender, and parental myopia as covariates. Spearman’s rank correlation coefficient test was performed to examine the associations between axial length and cylindrical components.

## 3. Results

### 3.1. Demographic Characteristics and Refractive Status

The mean ± SD age of the 418 schoolchildren who completed the vision screening and returned the questionnaire was 9.43 ± 0.93 years, of whom 55.9% were male. The average degree of astigmatism was 0.81 ± 0.72 DC, which was similar across ages (one-way ANOVA, F(3414) = 0.97, *p* = 0.41) and genders (unpaired *t*-test, *t*(416) = −0.22, *p* = 0.83). The average axial length and spherical-equivalent error were 23.71 ± 1.03 mm and −2.00 ± 1.51 D, respectively. Both parameters increased significantly with age (one-way ANOVA, F(3414) ≥ 4.34, *p* ≤ 0.005). Axial length was also longer in boys than in girls (unpaired *t*-test, *t*(416) = 5.87, *p* < 0.001), but no significant difference in SER was found between genders (unpaired *t*-test, *t*(416) = −0.54, *p* = 0.592).

### 3.2. Prevalence of Astigmatism

As shown in Table 1, the overall prevalence of astigmatism (Cyl ≥ 0.75 D) was 46.5% (95% CI, 41.7–61.4%), which was similar across age groups (Chi-squared test, χ^2^ (3) = 2.94, *p* = 0.40) and genders (Chi-squared test, χ^2^ (1) = 0.22, *p* = 0.64). For better comparison with other studies adopting other definitions of astigmatism, we also presented the prevalence of astigmatism with Cyl ≥ 0.50 D and ≥1.00 D as the criteria, which were 76.3% (72.0–80.4%) and 28.9% (24.6–33.5%), respectively. Neither age nor gender had a significant effect on the prevalence of astigmatism for either measurement (Chi-squared test, *p* ≥ 0.28).

Figure 1 presents the proportion of astigmatic subtypes according to the astigmatic axis. WTR astigmatism was predominant (66.1% to 82.43%), followed by oblique astigmatism (16.22% to 30.65%) and ATR astigmatism (1.35% to 3.85%). There were no significant differences in the proportion of astigmatism subtypes across age groups (Chi-squared test, χ^2^ (6) = 5.51, *p* = 0.48) or genders (χ^2^ (2) = 1.97, *p* = 0.37).

### 3.3. Astigmats vs. Non-Astigmats

We further divided schoolchildren into astigmatic (Cyl ≥ 0.75 D) and non-astigmatic groups (Cyl < 0.75 D) and compared their SER, axial length, and visual habits. As shown in Table 2, while astigmatic and non-astigmatic groups did not differ in age (unpaired *t*-test, *t*(416) = 0.90, *p* = 0.93) or proportion of gender (Chi-squared test, χ^2^ (1) ≤ 2.77, *p* ≥ 0.10), astigmatic children were 0.50 D more myopic and had 0.27 mm longer axial length than non-astigmatic children (unpaired *t*-test, *t*(416) = −3.93, *p* < 0.001, *t*(416) = 2.69, *p* < 0.001). Astigmatic children also spent about 0.3 h/day less time outdoors than non-astigmatic children (95% CI, 1.24–1.60 h/day vs. 1.52–1.96 h/day, unpaired *t*-test, *t*(416) = −2.10, *p* = 0.04). There were no significant differences in non-digital and handheld digital screen time between the two groups (unpaired *t*-test, *t*(416) ≤ 0.18, *p* ≥ 0.24).

### 3.4. Multiple Linear Regression Analyses

Multiple regression analyses were performed to determine how cylindrical power, and *J*0 and *J*45 astigmatic components were related to axial length and visual habits after controlling for age, gender, and parental myopia. The two models for cylindrical power and *J*0 astigmatic component were statistically significant (Table 3, F(7333) = 3.37, and F(7331) = 3.99, *p* < 0.001), with an adjusted R^2^ of 0.046 and 0.058, respectively. In both models, only axial length contributed significantly (β = 0.18 and β = 0.10, *p* < 0.001). According to the regression coefficients, children increased 0.18 D of cylindrical error and 0.10 D of *J*0 astigmatism for every 1 mm increase in axial length. Neither time spent on outdoor activities nor on near work (including the use of both non-digital materials and handheld digital devices) were associated with cylindrical error (*p* > 0.05) and *J*0 astigmatism (*p* > 0.05). On the other hand, the model for predicting *J*45 astigmatism was not significant (F(7331) = 1.52, *p* = 0.16), with an adjusted R^2^ of 0.011.

## 4. Discussion

The two key findings in this study were: (1) 46.5% of Hong Kong schoolchildren aged 8–11 years had refractive astigmatism ≥0.75 D following several months of lockdown during the COVID pandemic; (2) astigmatic schoolchildren were more myopic and had a longer axial length than non-astigmatic children.

Figure 2 plots the data from this study and those from ten epidemiological studies conducted on Asian Chinese children from 2000 to 2018 by stratifying them into three columns according to their definitions of astigmatism [6,7,8,9,10,11,12,20,21,22]. It clearly demonstrates that the prevalence of astigmatism after some months of COVID restrictions is higher than before COVID, regardless of how astigmatism is defined.

Only two of these ten epidemiological studies were conducted on Hong Kong Chinese schoolchildren. The most recent one was by Choi et al., who surveyed 1075 Hong Kong schoolchildren (mean age: 9.95 ± 0.97 years) in 8 local primary schools from June 2015 to February 2016 [12]. Their study design was similar to the current study, and the instrument used for refractive-error measurement was identical to ours. While Choi et al. did not report the prevalence of astigmatism, we obtained the astigmatism data through personal communication. By applying a definition of Cyl ≥0.50 D, ≥0.75 D, and ≥1.00 D, the prevalence rates were 59.4% (CI: 56.4–62.4%), 35.0% (CI: 32.1–38.0%), and 20.2% (CI: 17.8–22.7%), respectively, significantly lower than ours by 1.28 to 1.43 folds. In another study conducted in Hong Kong, the mean age was similar to that of the current study (9.33 years old, CI, 9.11–9.45), but the prevalence of astigmatism ≥1.00 DC was only 18.1% [22], which is 1.6-fold lower than our studied population (28.9%) but comparable to the data of Choi et al. A similar trend was observed in another study conducted on Singapore Chinese students aged 7 to 9 years [11], which showed that regardless of how astigmatism was defined, the prevalence of astigmatism was lower than the current study by 1.61-fold for Cyl ≥0.50 D (47.5% vs. 76.3%), 1.64-fold for Cyl ≥0.75 D (28.4% vs. 46.5%), and 1.49-fold for Cyl ≥1.00 D (19.4% vs. 28.9%). A lower prevalence of astigmatism was also noted in six pre-COVID studies conducted across different regions in China. The most dramatic difference from the current study was noted in the results from a study performed in Chongqing on children aged 8 to 11 years, in whom the prevalence was only 7.84% (CI: 4.89–10.79%) to 14.16% (CI: 10.45–17.87%) for Cyl ≥0.50 D and 2.19% (CI: 0.58–3.80%) to 5.60% (CI: 3.15–8.05%) for Cyl ≥1.00 D [21]. A possible explanation for such a considerable disparity may be that, unlike the dense urban community of Hong Kong, their study was conducted in a suburban area, where not only astigmatism but also the prevalence of myopia (13.75%) was much lower than in other rural areas of China. Of the other five epidemiological studies conducted in China, the highest astigmatism prevalence (≥0.75 DC) reported was from the one conducted in Shandong, ranging from 31.8% (CI: 28.5–35.2%) to 34% (CI: 30.1–38.0%), in children aged 8 to 11 years [8]. In Guangzhou [7], Beijing [6], Henan [9], and Shenzhen [10], the prevalence of astigmatism (≥0.75 DC) ranged from 9.5% to 26.3% (no CI was reported).

However, a study conducted in Taiwan showed a slightly higher prevalence of astigmatism (≥1.00 D) than the current study. The authors reported that 32.90% (no CI was reported) of Taiwanese children aged 8.97 ± 1.41 years were astigmatic [20], 1.14-fold higher than the current study. However, it should be noted that the Taiwan study only included one primary school, and their reported prevalence was 1.79-fold higher than an earlier study on a larger-scale epidemiological study in Taiwan that had reported only 18.4% of schoolchildren to be astigmatic (≥1.00 DC) [23].

For earlier astigmatism prevalence rates in Chinese children, one can refer to a recent meta-analysis that included 41 epidemiological studies conducted from 1983 to 2017 (total *n* = 1,051,784) [13]. The pooled prevalence of astigmatism was 16.5% (CI: 12.3–21.8%), 1.75 folds lower than the current study. However, the meta-analysis results should be interpreted with caution because the differences in definitions of astigmatism and geographic regions, as discussed above, have a significant effect on astigmatism prevalence.

Compared to previous epidemiological studies conducted on the prevalence of astigmatism in the Asian Chinese populations before COVID, an increase in prevalence was observed among Hong Kong Chinese schoolchildren during COVID. Although cycloplegic agents were not instilled prior to measuring refractive status through open-field autorefraction, it has been reported that ocular accommodation during refraction measurement has only a minimal effect on astigmatic readings [24]. It should be noted that there have been no apparent changes in Hong Kong’s educational system in the past few years, except that face-to-face teaching was suspended over certain periods to contain the COVID outbreaks. Recently, Zhang et al. reported that Hong Kong children spent 2.8 times more hours on digital devices, but three times less participating in outdoor activities during the school suspension period [14]. Similarly, a study conducted in Shanghai found that children spent more time on digital screens and engaged in fewer outdoor activities during COVID [25]. The current study also revealed that compared to non-astigmatic children, astigmatic children tended to spend 0.32 h/day less on outdoor activities. Whether the development of astigmatism in schoolchildren is triggered by exogenous visual signals due to the change in lifestyle warrants further study.

Another possible explanation for the high astigmatism prevalence is the association of astigmatism with axial myopic eye growth as a by-product [26,27,28]. Although it is unclear whether myopia had increased in our population during COVID because of the cross-sectional design and non-cycloplegic refraction, emerging evidence from studies in the Asian Chinese population indicates there has been escalated myopia progression during the COVID pandemic. A longitudinal study in Hebei, China, reported a significant myopia shift of school-age children seven months following the COVID outbreak (from January 2020 to August 2020) by −0.133 D/month, which was considerably more rapid than the myopia progression rate before COVID (−0.047 D/month, from July 2019 to January 2020) [25]. A large cross-sectional epidemiological study conducted in Shenzhen (a Chinese city adjacent to Hong Kong) also found a significant surge in the prevalence of myopia (SE <−0.50 D), from 46.9% in 2019 to 50.5% in 2020, in schoolchildren aged 7 to 12 years [16]. Compared with data acquired in the past five years, myopia prevalence in school children in Feicheng, China, was found to be increased substantially by 1.4 to 3-folds, with the averaged SER becoming more negative by approximately −0.30 D during COVID [15]. It has been proposed that the substantial myopia shift during the pandemic was due to increased digital screen time [25], although more reading hours could also be a potential myopigenic factor [14]. The reduced outdoor time, associated with reduced retinal dopamine secretion [29] and vitamin D formation [30], could be another possible explanation for the increased myopia prevalence [31,32]. The results of the current study support the idea that increased axial myopia during COVID might be linked to the high prevalence of astigmatism. We found that compared with the non-astigmatic children, astigmatic children had a longer axial length (0.27 mm longer) and more negative SER (−0.53 D more negative). In addition, multiple linear regression analyses also indicated significant relationships between axial length with cylindrical error and *J*0 astigmatic component: a millimeter increase in axial length was accompanied by about a 0.20 D increase in WTR astigmatism (i.e., 0.10 D in *J*0 astigmatism). Such a relationship agrees with other epidemiological studies, supporting the coexistence of astigmatism (not causation), specifically WTR astigmatism, with myopia [2,11,26,27,33]. Further studies are warranted to ascertain the relationship between astigmatism and myopia.

This study revealed that the prevalence of astigmatism in Hong Kong schoolchildren was considerably higher during the COVID pandemic and was associated with axial myopia. However, there are several limitations in our study design. First, because of the tight teaching schedule after school re-opening, the cycloplegic agent was not administered to avoid the lengthy process disturbing the children’s regular school learning. To relax ocular accommodation, an open-field autorefractor was used to measure refractive errors by asking participants to fixate at an external fixation target 6 m away from the eye. However, it should be noted that cylinder refractive measurements are shown to generate similar readings (mean difference of −0.08 ± 0.13 D for *J*0 and −0.01 ± 0.09 D for *J*45) under cycloplegic or non-cycloplegic conditions [24]. Additionally, Kuo et al. [34] who used the same model of open-field auto-refractor as ours (Shin-Nippon, NVision-K 5001, Tokyo, Japan), reported no significant difference between cycloplegic and non-cycloplegic *J*0 and *J*45 astigmatism in children aged 6–11 years. Second, although we followed the Government’s guidelines on social distancing and disinfected the instruments regularly, the participation rate was somewhat low (69%), probably because some parents were concerned about potential cross-infection with the COVID virus. Third, while the prevalence of astigmatism reported in this study was higher than in most previous studies conducted on the Asian Chinese population, there was no control group for a direct comparison. Thus, it requires more longitudinal studies to understand better how the lifestyle change during COVID affects refractive error development. Lastly, using a questionnaire to report near-work and outdoor activities is subject to recall bias. Other working distance and light intensity measurement gadgets could provide more information on children’s visual habits. Nevertheless, because of the converging evidence of increasing myopia and astigmatism during COVID, eye care practitioners and parents need to be aware of the potential changes in the visual status of children. More investigations are also needed to understand the underlying mechanism of accelerating myopia and astigmatism development.

## Figures and Tables

**Figure 1 children-09-00919-f001:**
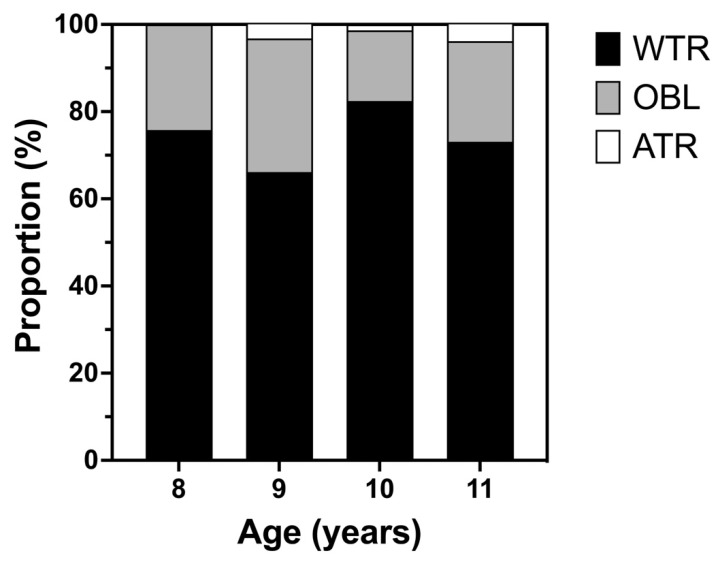
Proportions of With-The-Rule (WTR), Against-The-Rule (ATR), and Oblique (OBL) astigmatism across ages.

**Figure 2 children-09-00919-f002:**
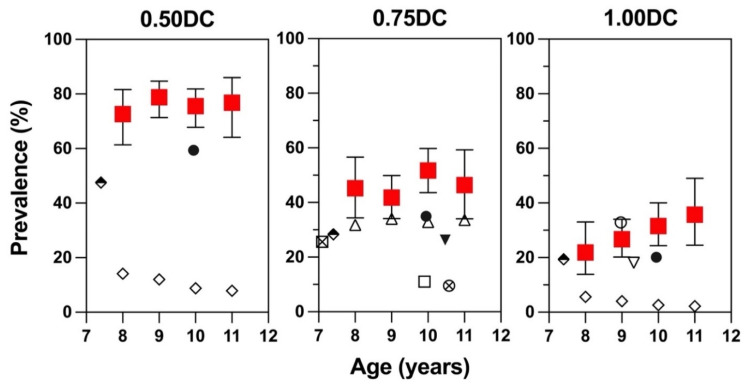
Comparison of prevalence rates of astigmatism in this study (red filled square 
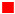
) with control data in Hong Kong ⚫ [12]; ▽ [22] and other Asian Chinese populations, including Singapore ⬘ [11], Taiwan ○ [20], and Mainland China ⊗ [6]; ▼ [7]; △ [8]; ⊠ [9]; ☐ [10]; ◇ [21]. Only recent studies conducted on Asian Chinese population that consistently used either right eye or left eye to report the prevalence of astigmatism were included. Prevalence rates of various studies were stratified into three columns according to their definitions of astigmatism (Cyl ≥ 0.50, 0.75, and 1.00 DC). Note that data from Choi et al. 2017 [12] were not published but obtained via personal communication with the authors.

**Table 1 children-09-00919-t001:** Comparisons of prevalence of different definitions of astigmatism across age and gender.

Definitions		*n*	Astigmatism ≥ 0.50 D		Astigmatism ≥ 0.75 D		Astigmatism ≥ 1.00 D	
			Prevalence (95% CI)	*p* Values	Prevalence (95% CI)	*p* Values	Prevalence (95% CI)	*p* Values
Total		418	76.3 (72.0–80.4)		46.5 (41.7–61.4)		28.9 (24.6–33.5)	
Age	8	73	72.6 (61.4–81.6)	0.77	45.2 (34.4–56.6)	0.40	21.9 (13.9–32.8)	0.28
	9	146	78.8 (71.4–84.7)		41.8 (34.1–49.9)		26.7 (20.2–34.4)	
	10	143	75.5 (67.8–81.9)		51.7 (43.6–59.8)		31.5 (24.4–39.5)	
	11	56	76.8 (64.1–86.0)		46.4 (34.0–59.3)		35.7 (24.5–48.8)	
Gender	Male	234	76.5 (70.6–81.5)	0.92	47.4 (41.1–53.8)	0.64	29.1 (23.6–35.2)	0.86
	Female	184	76.1 (69.5–81.7)		45.1 (38.1–52.3)		28.3 (22.3–35.2)	

**Table 2 children-09-00919-t002:** Comparison of demographic information, myopia, and visual habits between astigmatic and non-astigmatic children. Bold values denote statistical significance (*p* < 0.05).

	Astigmats (*n* = 195)	Non-Astigmats (*n* = 223)	*p* Values
Age (years)	9.48 ± 0.93	9.40 ± 0.93	0.927
Gender			
Males (%)	57.2	54.9	0.636
Spherical-equivalent Error (D)	−1.88 ± 1.67	−1.35 ± 1.04	**<0.001**
Axial Length (mm)	23.85 ± 1.18	23.58 ± 0.86	**<0.001**
Reading Time (hour)	1.33 ± 0.87	1.32 ± 0.97	0.994
Screen Time (hour)	2.50 ± 1.82	2.47 ± 2.15	0.242
Outdoor Activities Time (hour)	1.42 ± 1.30	1.74 ± 1.67	**0.044**

**Table 3 children-09-00919-t003:** Multiple linear regression models for prediction of cylindrical power, *J*0, and *J*45 astigmatism.

	Beta	*p*-Value
**Cylindrical Error**		
Axial length	0.181	**<0.001**
Reading hours	0.003	0.946
Screen hours	−0.011	0.584
Outdoor hours	−0.017	0.498
***J*0 Astigmatism**		
Axial length	0.099	**<0.001**
Reading hours	−0.013	0.543
Screen hours	−0.016	0.123
Outdoor hours	−0.007	0.579
***J*45 Astigmatism**		
Axial length	0.002	0.872
Reading hours	−0.01	0.443
Screen hours	0.017	**0.013**
Outdoor hours	−0.011	0.161

Age, gender, and parental myopia were adjusted as covariates. Bold values denote statistical significance at the *p* < 0.05 level.

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
