# Peer review of "High Prevalence of Astigmatism in Children after School Suspension during the COVID-19 Pandemic Is Associated with Axial Elongation"

_children, 2022, doi:10.3390/children9060919_

Round 1

Reviewer 1 Report

Congratulation for the article

Author Response

Thanks!

Reviewer 2 Report

Interesting manuscript. The work concerns the assessment of the progression of astigmatism in children during a pandemic. It clearly shows that the reduction of the time spent in the open air and the increase in the time of working at the near distance not only lengthens the eyeball but also influence astigmatism value.

Author Response

Thanks!

Reviewer 3 Report

The authors have analyzed the occurrence of astigmatism and myopia in 418 ethnically Chinese children at three local primary schools in Hong Kong from October 2020 to December 2020, during the lockdown period, when kids were forced to stay home and work at the screen to follow lessons an perform their duties. They found a higher prevalence then expected based on bibliographich data of both astigmatism and myopia. Astigmatism affects the cornea and is due to an unequal curvature along its axes. Myopia follows an abnormal elongation of the longitudinal axis of the eye globe. This could put a strain on the cornea, and change its shape. However, a clear correlation between astigmatism and myopia has not been established. Data here presented hint at such correlation (tables 2 and 3), however the time period to which it refers is quite short, and the correlation still weak. There is no evident clustering of subjects on a hypothetical 3D graph taking into account astigmatism, myopia and outdoor time. A fourth element that could have been considered, at least in the discussion, is vitamin D, which is known to strongly contribute to eye health, and myopia in particular. Nonetheless, the data here reported suggest that forcing kids at home and in front of screens, and preventing them from spending more time outside, also for a limited time period, might have a deleterious effect on their eye growth, inducing visual aberrations.

Please note that supplementary figure 2 is identical to figure 2 in the manuscript. Please, correct this mistake.

Reviewer 4 Report

Thank you for allowing me to review this paper

1.       The effect of Covid on refraction is of interest. Why did you focus on astigmatism and not the sphere?

2.       The period was 02-07.2020- will this time be enough for effect?

3.       What was the control? I would expect the control should be children examined the year before

4.       . Due to the increase in refractive-error development- this is a bias and a limitation- you need a control group, not a "literature group." You also state that "in the Asian Chinese population, the prevalence of astigmatism (≥0.75DC) varies across geographic regions, "so a control group is needed here

5.       I see Ref 16 had a control group to show an increase in astigmatism---so what is new In the current study? You had a larger sample but no control

6.        "Due to the increase in refractive-error development," this was not proven as you had only one exam- and no control group.

7.       With which version of IOLMaster did you measure axial length? 500 or 700

8.       Can astigmatism be corrected with head tilt?

9.       Please add the questionnaire as a supplemental

10.   One eye per patient- excellent design

11.   The mean degree of astigmatism was 0.81±0.72DC—this seems normal, no?

12.   The mean degree of astigmatism was 0.81±0.72DC—was this based on autoamtimc refraction or optians testing? My refractor is always over minus astigmatism- this may be due to the pupil size effect- were the children examined under a cycloplegic condition?

13.   Table 1 seems redundant as all this data is on the text, no?

14.   I don't understand table 1 – which groups are compared? Is this an ANOVA test?

15.   Astigmatic children also spent about 0.3 hours/day less time outdoors than non-astigmatic 167 children (unpaired t-test)---this is an excellent finding, so only 20 minutes maybe relate to astigmatism? We don't know, as we don't have astigmatism before the covid – but there is a signal here.

16.   I think you need to compare the mean 95CI difference between the groups in terms of outdoor Activities Time (hour)

17.   Did you control for multiple comparions0- I am afraid that 0.044 would not be statistical  significant  anymore if you perform correction

18.    Table 4- missing is the sphere or SE in the models of linear regression

19.   How can you explain that the Outdoor hours were significant in  the j45 and not J0

20.   What is the difference between J 45 and J0 ( for the non-optician reader)

21.   I  am not sure what is new here- correlation between the cylinder and axial length?

22.   Figure 2- don't you need to run a power analysis? Or meta-analysis to control for N of each study?

23.   how can you compare these studies as they are from a different region in Asia and has a mixed population – I think an excellent meta-analysis is warranted her

24.   can you explain figure 2 in the legend? Help the reader understand your point?

25.   "Another possible explanation for the surge in astigmatism" you did not show a surge- I will rephrase and be more careful (ross- sectional design and non-cycloplegic refraction)

26.   This study revealed that the prevalence of astigmatism in Hong Kong schoolchildren 277 was considerably higher during the COVID pandemic and was associated with axial my- 278 opia---no data is given to support this. You don't have control.

27.    

28.   Overall, the design here won't allow us to draw any meaningful conclusion (no control, no longitudinal examination); however, observing a possible connection between outdoor activity and astigmatism and SE is important. I think it is wise that researchers and doctors read about these connections. However, the linear regression did not confirm this, which showed that only screen time predicts the J45.

29
